JCB Journal of Cell Biology

# REPORT

# Endocytic myosin-1 is a force-insensitive, power-generating motor

Ross T.A. Pedersen[1]*  , Aaron Snoberger[2]*  , Serapion Pyrpassopoulos[2]  , Daniel Safer[2]  , David G. Drubin[1]  , and E. Michael Ostap[2]

**Myosins are required for clathrin-mediated endocytosis, but their precise molecular roles in this process are not known. This is, in part, because the biophysical properties of the relevant motors have not been investigated. Myosins have diverse mechanochemical activities, ranging from powerful contractility against mechanical loads to force-sensitive anchoring. To better understand the essential molecular contribution of myosin to endocytosis, we studied the in vitro force-dependent kinetics of the *Saccharomyces cerevisiae* endocytic type I myosin called Myo5, a motor whose role in clathrin-mediated endocytosis has been meticulously studied in vivo. We report that Myo5 is a low-duty-ratio motor that is activated ~10-fold by phosphorylation and that its working stroke and actin-detachment kinetics are relatively force-insensitive. Strikingly, the in vitro mechanochemistry of Myo5 is more like that of cardiac myosin than that of slow anchoring myosin-1s found on endosomal membranes. We, therefore, propose that Myo5 generates power to augment actin assembly-based forces during endocytosis in cells.**

## Introduction

During clathrin-mediated endocytosis (CME), the plasma membrane invaginates and undergoes scission to become a cytoplasmic vesicle. Coat proteins like clathrin can deform membranes under low tension (Dannhauser and Ungewickell, 2012; Busch et al., 2015; Cail et al., 2022), but when bending is resisted by membrane tension (Hassinger et al., 2017), the actin cytoskeleton drives membrane invagination (Boulant et al., 2011; Kaplan et al., 2022). In yeasts, including *Saccharomyces cerevisiae* and *Schizosaccharomyces pombe*, turgor pressure opposes plasma membrane invagination, so actin is required at every CME site (Aghamohammadzadeh and Ayscough, 2009; Basu et al., 2014).

The actin cytoskeleton can produce pushing and pulling force, both of which are required for CME in *S. cerevisiae* (Sun et al., 2006). When actin filament ends grow against a surface, they push the surface forward (Mogilner and Oster, 1996, 2003). During CME, actin filaments, bound by coat proteins, grow against the plasma membrane to drive invagination (Picco et al., 2015; Kaksonen et al., 2005, 2003; Skruzny et al., 2012, Fig. 1). Modeling of the homologous CME machinery in mammalian cells demonstrated that such actin networks generate sufficient power for CME (Akamatsu et al., 2020), but whether actin assembly alone can overcome turgor pressure in yeast cells is debated (Nickaeen et al., 2019; Carlsson, 2018).

Additional power may be provided by myosins, which generate tension on actin filaments. The myosins critical for CME—Myo3 and Myo5 in budding yeast and Myo1e in vertebrates—are type I myosins (Geli and Riezman, 1996; Cheng et al., 2012; Krendel et al., 2007). Some type I myosins are suited to generate power—i.e., they carry out mechanical work over time by consuming ATP to execute a power stroke. Other type I myosins are suited to serve as force-sensitive anchors in that mechanical load locks them in a low energy–requiring, tension-maintaining state (Greenberg and Ostap, 2013). The possible roles of type I myosins in CME depend on whether endocytic myosins are power generators or force-sensitive anchors.

If endocytic type I myosins are acutely force sensitive, they might organize the actin filaments of the endocytic actin network, while if they are less force sensitive, they could power plasma membrane invagination (Evangelista et al., 2000; Fig. 1). Myosin-1 motors form a ring at the base of CME sites, where the invaginated membrane meets the plasma membrane (Mund et al., 2018; Fig. 1). Yeast type I myosins serve at least one organizational function as a membrane anchor for the actin assembly machinery, a function associated with the non-motor tail of the molecules (Lewellyn et al., 2015), but motor activity is required in addition to membrane anchorage (Pedersen and

[1]Department of Molecular and Cell Biology, University of California, Berkeley, Berkeley, CA, USA; [2]Pennsylvania Muscle Institute, Perelman School of Medicine, University of Pennsylvania, Philadelphia, PA, USA.

Correspondence to David G. Drubin: drubin@berkeley.edu; E. Michael Ostap: ostap@mail.med.upenn.edu

*R.T.A. Pedersen and A. Snoberger contribution equally to this paper. R.T.A. Pedersen's current affiliation is Department of Embryology, Carnegie Institution for Science, Baltimore, MD, USA.

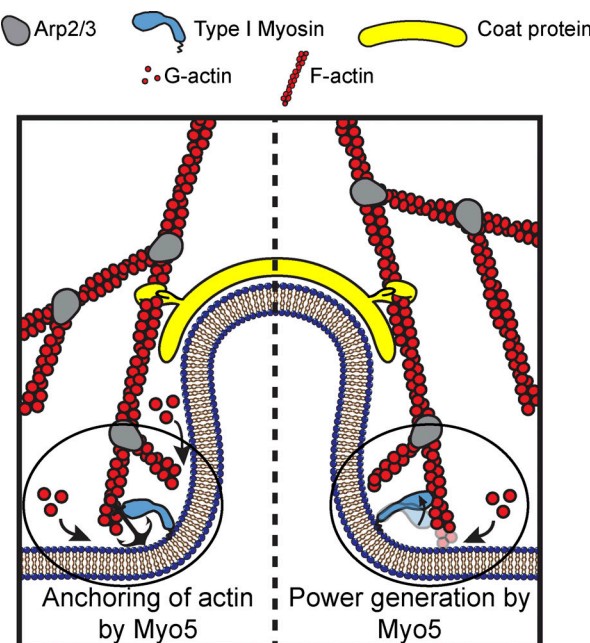

**Figure 1. Models for the functions of actin assembly and myosin activity during membrane deformation for clathrin-mediated endocytosis.** Cartoon diagram illustrating the organization of actin filaments and Myo5 molecules at endocytic sites. Actin filaments are bound by coat proteins at the tip of the growing membrane invagination and oriented with their growing ends toward the plasma membrane, powering membrane invagination. The type I myosin Myo5 could either anchor the actin network in a favorable orientation (left) or provide an assisting force (right).

Drubin, 2019). If endocytic myosin-1s are force-sensitive anchors, they may serve a further organizational role by holding growing filaments in an optimal orientation for force generation (Fig. 1, left). If the myosins are power-generating motors, they may pull actin filament ends away from the plasma membrane, deepening the plasma membrane invagination and creating space for monomer addition and filament elongation (Fig. 1, right), a model supported by the observation that the actin assembly rate at CME sites depends on type I myosin motors in a dose-dependent manner (Manenschijn et al., 2019).

To distinguish between these possibilities, we measured the force sensitivity of the endocytic myosin Myo5 (not to be confused with the vertebrate type V myosin). Myo5 is insensitive to resistive force compared to related myosins. We, therefore, propose that Myo5 actively powers CME. Because actin and myosin collaborate in a variety of membrane remodeling processes, we expect that these results will be instructive beyond CME.

## Results and discussion

### Heavy chain phosphorylation activates Myo5 ATPase activity

To determine Myo5 force sensitivity, we first needed to measure its unloaded kinetics. We purified a Myo5 construct containing the motor and lever domains from *S. cerevisiae* (Fig. 2 A). Because phosphorylation of Myo5 at the TEDS site is required for most CME events and is thought to regulate Myo5's motor activity (Grosshans et al., 2006; Sun et al.,

2006; Bement and Mooseker, 1995), we purified a phosphorylated version and an unphosphorylated version of the protein (see Materials and methods). The p21-activated kinase was used to phosphorylate the myosin at the TEDS site (S357), as determined by control experiments with an S357A mutant (Fig. S1). The phosphorylation state of preparations was judged to be uniform when ATP-induced actoMyo5 dissociation transients were well fit by single exponential functions (see below). The yeast light chain for Myo5, calmodulin (Cmd1, Geli et al., 1998), was purified from *E. coli* and included in excess in all experiments (Fig. 2 A).

We measured the steady-state actin-activated ATPase activities of phosphorylated and unphosphorylated Myo5 using the NADH-coupled assay (De La Cruz and Ostap, 2009) in the presence of 0–80 µM phalloidin-stabilized actin filaments. Unphosphorylated Myo5 ATPase activity was largely insensitive to actin filaments: the ATPase rate at 0 µM actin was 0.14 s$^{-1}$, while the maximum ATPase rate measured was 0.39 s$^{-1}$ at 40 µM actin. (Fig. 2 B). Phosphorylation activated Myo5 ATPase activity by ∼10-fold (Fig. 2 B). The actin concentration dependence of the phosphorylated Myo5 ATPase rate ($k_{obs}$) was well fit by:

$$k_{obs} = v_o \frac{V_{max}[Actin]}{K_{ATPase} + [Actin]}. \qquad (1)$$

From the fit, the actin concentration at half-maximum of the ATPase rate ($K_{ATPase}$) was determined to be 5.1 ± 0.88 µM, and the maximum ATPase rate ($V_{max}$) was found to be 3.3 ± 0.15 s$^{-1}$ (Fig. 2 B; Table 1).

### ATP binding and ADP release are non-rate limiting for Myo5 ATPase activity

Resistive force impacts the rate of myosin detachment from actin and two biochemical transitions, ADP release and subsequent ATP binding, determine the detachment rate. Therefore, we used stopped-flow kinetics to measure ADP release from (Fig. 2 C, $k_{+5}'$) and ATP binding to (Fig. 2 C, $K_1'$ and $k_{+2}'$) actoMyo5.

We found that yeast Myo5 does not quench the fluorescence of actin labeled at cys-374 with pyrene iodoacetamide, which is the probe most used to measure myosin binding (De La Cruz and Ostap, 2009). Thus, we measured actoMyo5 detachment by monitoring light scattering, which decreases as myosin unbinds actin filaments.

To determine the rate constant for ATP binding, we mixed nucleotide-free actoMyo5 (100 nM) with varying concentrations of ATP and monitored 90° light scattering. Time courses followed single exponential functions (Fig. 2 D). For phosphorylated Myo5, the observed rates determined from the fits increased linearly with ATP concentration (Fig. 2 E). At concentrations of >1 mM ATP, the actomyosin complex dissociated within the response time of the instrument, precluding measurement. For unphosphorylated Myo5, the observed rates fit a rectangular hyperbola with increasing ATP concentration (Fig. 2 E).

$$AM + ATP \underset{}{\overset{K_1'}{\rightleftharpoons}} AM(ATP) \overset{k_2'}{\rightarrow} AM.ATP \overset{k_{diss}}{\rightarrow} A + M.ATP$$

Scheme 1

The mechanism was modeled as in Scheme 1 (De La Cruz and Ostap, 2009), where $K_1'$ is a rapid equilibrium binding step, $k_2'$ is

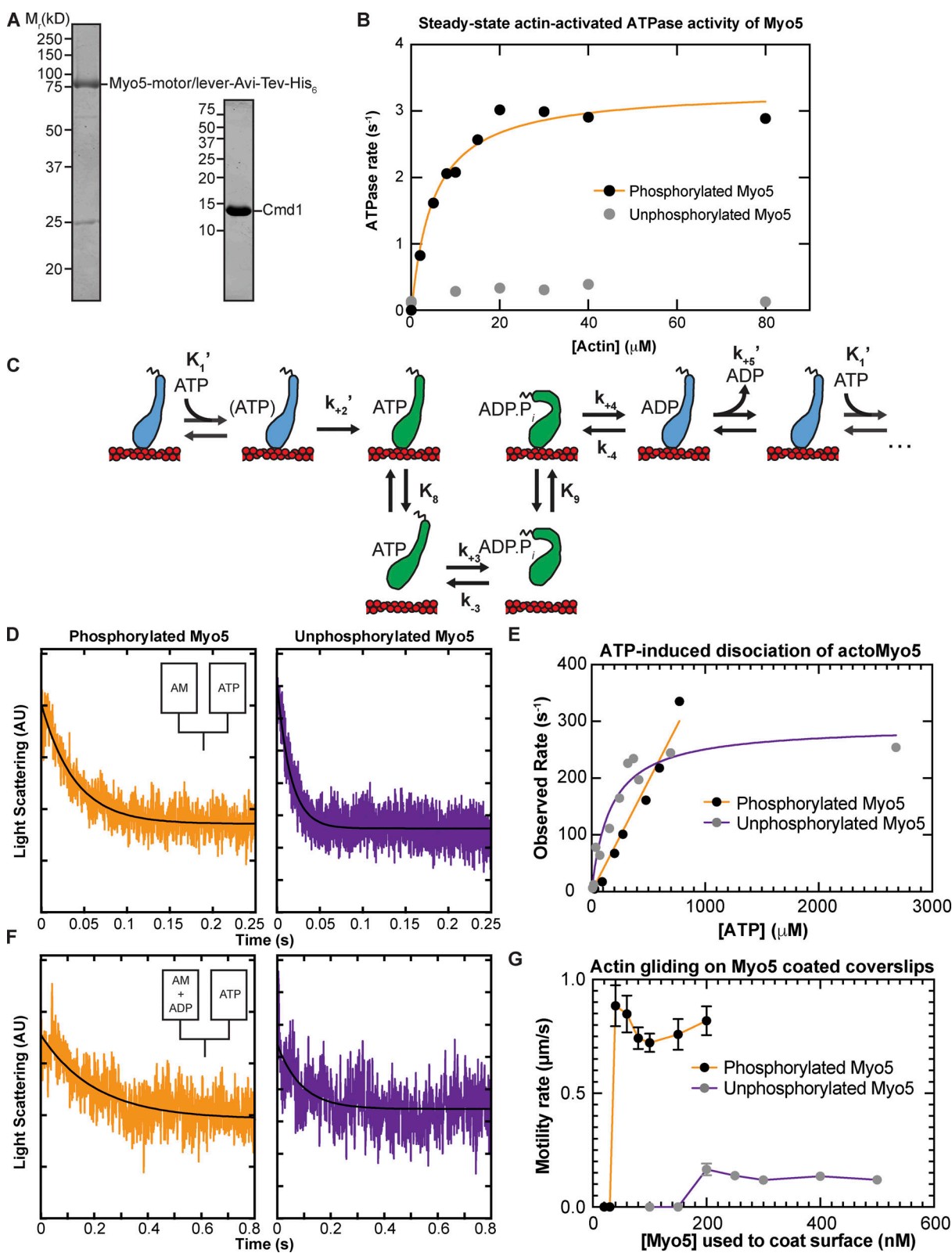

Figure 2. **In-solution, population biochemical characterization of Myo5. (A)** Coomassie-stained SDS-polyacrylamide gels showing example preparations of the purified Myo5 motor/lever construct and calmodulin (Cmd1, light chain) used in all experiments. **(B)** The actin concentration dependence of the steady-state ATPase activity of 100 nM unphosphorylated (gray circles) and phosphorylated Myo5 (black circles). Each data point represents the average of 6–7 time courses, which were 100 s each. The orange line is the best fit of the phosphorylated Myo5 data to a rectangular hyperbola. **(C)** Schematic pathway for the Myo5 ATPase cycle. Blue motors are in tightly bound conformations and green motors are weakly bound/unbound. **(D)** Example of light scattering transients reporting on ATP-induced dissociation of phosphorylated (left, $k_{obs}$ = 17 s$^{-1}$) and unphosphorylated (right, $k_{obs}$ = 64.1 s$^{-1}$) actoMyo5, obtained by mixing 100 nM actoMyo5 (AM) with 94 and 72 µM ATP, respectively, as shown in the inset schematic. The black line is the fit of a single exponential function to the data. **(E)** ATP

concentration dependence of dissociation of 100 nM unphosphorylated (gray circles) and phosphorylated actoMyo5 (black circles). Each data point represents 3–6 time courses averaged and fit to a single exponential decay function. The orange line is a linear best fit of the phosphorylated Myo5 data. The purple line is the best fit of the unphosphorylated Myo5 data to a rectangular hyperbola. **(F)** Example light scattering transients reporting ATP-induced dissociation of ADP-saturated phosphorylated (left) and unphosphorylated (right) actoMyo5, obtained by preincubating 200 nM actoMyo5 (AM) with 100 µM ADP, then mixing rapidly with 2.5 mM ATP, as shown in the inset schematic. The black line is the fit of a single exponential function to the data. **(G)** Velocity of actin filament gliding, measured at varying surface densities of Phospho-Myo5 (black circles, orange line) and unphosphorylated Myo5 (gray circles, purple line) in in vitro motility assays. Myosin concentrations indicate the quantity of protein incubated in the flow chamber before washing. Each data point represents the average velocity of 30–60 filaments, and the error bars are standard deviations. Source data are available for this figure: SourceData F2.

a rate-limiting isomerization to the AM.ATP state, and $k_{diss}$ is the rapid actin dissociation step. The apparent second-order rate constant for ATP binding to phosphorylated actoMyo5 was determined by a linear fit to the data ($K_1'k_2' = 0.39 \pm 0.017\ \mu m^{-1}\ s^{-1}$). The unphosphorylated actoMyo5 data were fit by

$$k_{obs} = \left[ \frac{K_1'[ATP]}{1 + K_1'[ATP]} \right] k_2' \qquad (2)$$

and the maximum rate of isomerization ($k_2' = 290 \pm 24\ s^{-1}$) and ATP affinity ($K_1' = 0.006 \pm 0.0016\ \mu M^{-1}$) were determined. The apparent second-order rate constant for ATP binding ($K_1'k_2'$) was determined from a linear fit of the observed rates below 100 µM ATP to be $1.1 \pm 0.28\ \mu M^{-1}\ s^{-1}$ (Table 1).

The rate constant for ADP dissociation ($k_{+5}'$) was measured by preincubating 100 µM ADP with 200 nM actoMyo5 and then rapidly mixing with 2.5 mM ATP, as shown in Scheme 2.

$$A.M.ADP \overset{k_5'}{\rightleftharpoons} A.M + ATP \rightleftharpoons A.M(ATP) \longrightarrow A + M.ATP$$

Scheme 2

When myosin active sites are saturated with ADP, the rate of ATP-induced dissociation of actomyosin is limited by ADP's slow dissociation. Light scattering transients were fitted by single exponential functions, yielding rates for ADP release for

phosphorylated actoMyo5 ($k_{+5}' = 74 \pm 2.0\ s^{-1}$) and for unphosphorylated actoMyo5 ($k_{+5}' = 107 \pm 5.9\ s^{-1}$; Fig. 2 F and Table 1). The signal-to-noise ratio of the fast light scattering transients is low, resulting in large uncertainties on these fits. However, these rates are substantially faster than the steady-state ATPase values but slower than the maximum rate of ATP-induced actomyosin dissociation. ADP release for actoMyo5 ADP is much faster than ADP release for vertebrate Myo1b and Myo1c (Greenberg et al., 2012; Lewis et al., 2006). It is more similar to the vertebrate endocytic myosin-1, Myo1e (El Mezgueldi et al., 2002). Because ADP release is rate limiting for detachment of Myo5 and Myo1e from actin, fast ADP release by these molecules means that the unloaded actin-attachment lifetimes for endocytic type I myosins are <15 ms. This property may make these motors particularly well-suited to function in dynamic actin networks like those at CME sites, where actin filaments elongate and "treadmill" into the cytoplasm (Kaksonen et al., 2003, 2005).

### Actin gliding is dependent on Myo5 phosphorylation state

Our results suggest that both phosphorylated and unphosphorylated Myo5 have low duty ratios (i.e., the motor spends a small fraction of its ATPase cycle bound to actin). Since ADP release limits the rate of phosphorylated Myo5 detachment from actin at saturating ATP ($k_{+5}' = 74 \pm 2.0\ s^{-1}$) and since we have measured the overall ATPase rate ($V_{max} = 3.3 \pm 0.15\ s^{-1}$), we can estimate the duty ratio as follows:

$$DutyRatio = \frac{\left( \frac{1}{k_{+5}'} \right)}{\left( \frac{1}{V_{max}} \right)} \qquad (3)$$

The calculated duty ratio of phosphorylated Myo5 is 0.045. Unphosphorylated Myo5 has a lower duty ratio (<0.004).

To assess the effect of phosphorylation on Myo5 motility, we performed in vitro motility assays at 1 mM ATP. Motors were attached site-specifically to coverslips coated with anti-His$_6$ antibody. Coverslips were incubated with a range of concentrations of phosphorylated and unphosphorylated Myo5, creating a titration series of surface densities. At low Myo5 surface densities (incubation with ≤30 nM phosphorylated Myo5, ≤150 nM unphosphorylated Myo5), actin filaments failed to bind the coverslip (Fig. 2 G; and Videos 1 and 2). At higher concentrations, phosphorylated Myo5 moved actin filaments at velocities ranging from 720 ± 40 nm/s (100 nM phosphorylated Myo5) to 880 ± 90 nm/s (40 nM; Fig. 2 G and Video 1). These gliding velocities are considerably higher than those reported by Sun et al., (2006), possibly reflecting differences in the phosphorylation state of the purified Myo5 protein (see below) or differences in other motility assay conditions, such as light chain availability. Higher (greater

Table 1. **Rate and equilibrium constants of the Myo5 ATPase cycle**

| | Phosphorylated Myo5 | Unphosphorylated Myo5 |
|---|---|---|
| **Steady-state actin-activated ATPase** | | |
| $V_{max}$ (s$^{-1}$) | 3.3 (±0.15) | ND |
| $K_{ATPase}$ (µM) | 5.1 (±0.88) | ND |
| **ATP binding** | | |
| $K_1'$ (µM$^{-1}$) | ND | 0.006 (±0.0016) |
| $k_2'$ (s$^{-1}$) | ≥335 | 290 (±24) |
| $K_1'k_2'$ (µM$^{-1}$ s$^{-1}$)[a] | 0.39 (±0.017)[b] | 1.1 (±0.28)[c] |
| **ADP release** | | |
| $k_{+5}'$ (s$^{-1}$) | 74 ± 2.0 | 107 (±5.9) |

Summary of rate and equilibrium constants measured for Myo5 in this study. Errors are standard errors of the fits.
[a]Determined from a linear fit of the unbinding rates.
[b]Linear fit of all data for phosphorylated Myo5 in Fig. 2 E.
[c]Linear fit of observed rates below 100 µM ATP for unphosphorylated Myo5 in Fig. 2 E. ND: Not determined.

than fivefold) surface densities of unphosphorylated Myo5 were required to achieve smooth motility, but this motility occurred at a substantially slower speed, ~120 nm/s (Fig. 2 G and Video 2). While it is possible that residual phosphorylated Myo5 in the unphosphorylated preparation contributed to this motility, Sun et al., 2006 similarly reported that Myo5 harboring TEDS site mutations moved actin filaments much more slowly. The slower actin gliding speed for unphosphorylated myosin was unexpected given the similar rates of ADP release between phosphorylated and unphosphorylated Myo5 (Table 1). It is possible that our kinetics experiments have not determined the rate-limiting step for detachment, but it is more likely that the motility of the unphosphorylated myosin is limited by the slow attachment rate of the motor (Stewart et al., 2021), as suggested by the slow actin-activated ATPase rate. The activation of Myo5 motility by phosphorylation could explain why fast, cargo-induced endocytosis, which involves rapid and dynamic actin turnover, requires phosphorylated Myo5, while slower constitutive endocytosis does not (Grosshans et al., 2006).

## Myo5's working stroke comprises two substeps that are consistent with unloaded kinetic measurements

The kinetics of actin attachment and mechanics of single myosin molecules were measured by optical trapping (Woody et al., 2018; Snoberger et al., 2021). We used the three-bead optical trapping geometry in which a biotinylated actin filament is held between two laser-trapped polystyrene beads coated with neutravidin, creating a bead-actin-bead dumbbell (Fig. 3 A). Dumbbells were lowered onto pedestal beads that were sparsely coated with phosphorylated Myo5-His$_9$ bound to a surface-adsorbed anti-His$_6$ tag antibody. The positions of trapped beads were detected, and single actomyosin binding events were identified by a decrease in the covariance of the bead positions (Fig. 3, B–D).

Traces acquired at 1, 10, and 1,000 µM ATP reveal clear displacements and drops in covariance during actomyosin binding events. Event durations decreased with increasing ATP concentrations (Fig. 3, B–D, blue lines).

The myosin-1 working stroke occurs in two discrete substeps, with the first substep occurring with phosphate release and the second with ADP release (Jontes et al., 1995; Veigel et al., 1999, Fig. 3 E). The substeps can be characterized in optical trapping assays by ensemble averaging single interactions (Veigel et al., 1999; Chen et al., 2012; Laakso et al., 2008), where the detected events are aligned at their beginnings and forward-averaged in time (Fig. 3, F–H, left), or aligned at their ends and reverse-averaged in time (Fig. 3, F–H, right).

Ensemble averages of Myo5 interactions showed a two-step working stroke at the three ATP concentrations but the step size was most accurately resolved at 10 µM ATP (see Materials and methods). In this condition, an initial substep of 4.8 nm was followed by a second substep of 0.2 nm (Fig. 3 G). We determined the lifetimes of the substeps by fitting the ensemble averages with single exponential functions. At 1 µM ATP (Fig. 3 F, left trace), the measured rate (>30 s$^{-1}$) of the time-forward average was limited by the covariance smoothing window, but at 10 and 1,000 µM ATP (Fig. 3, G and H, left traces), the rates were 49 ± 1.6 and 50 ± 0.2 s$^{-1}$, respectively

(Fig. 3 K), which are similar to the measured ADP release rate ($k_{+5}'$, 74 ± 2.0 s$^{-1}$, Table 1) supporting the model that the transition from State 1 to State 2 accompanies ADP release.

The kinetics of time-reversed averages reveal the lifetime of State 2 (Fig. 3, F–H, right traces). Fitting single exponential functions to these traces reveals rates of 0.59 ± 0.003 and 7.34 ± 0.1 s$^{-1}$ at 1 and 10 µM ATP, respectively (Fig. 3 K). At 1,000 µM ATP, the observed rate (>187 s$^{-1}$) was limited by the covariance smoothing window (5.25 ms; Fig. 3 K). The observed rates at 1 and 10 µM ATP are consistent with the second-order rate constant for ATP binding of 0.39 ± 0.017 µM$^{-1}$ s$^{-1}$ measured by stopped-flow kinetics ($K_1'k_2'$, Table 1).

We determined the detachment rates of actomyosin events by plotting the cumulative frequency of individual attachment durations and fitting a single exponential function to the data by maximum likelihood estimation (MLE; Fig. 3 I). Data from 1 and 10 µM ATP were well fit by single exponentials with rates of 0.88 and 6.87 s$^{-1}$, respectively (Fig. 3, I–K). These rates match well with the observed rate of ATP binding (Table 1), as well as the fits for the reverse ensemble averages, indicating that at sub-saturating ATP (1 and 10 µM), detachment is limited by ATP binding (Fig. 3, I–K). Data from 1,000 µM ATP were best described as the sum of two exponentials, with the major rate of 67.8 s$^{-1}$ comprising 92.1% of the total, and a minor rate of 11.6 s$^{-1}$ comprising 7.9% of the total (Fig. 3, I and K). The major rate is consistent with both the observed ADP release rate and the measured forward ensemble average rates, indicating that at saturating ATP, ADP release limits detachment of actomyosin interactions (Fig. 3, J–K).

## Myo5 is a relatively force-insensitive motor

We measured how the actin detachment rate of Myo5 was affected by mechanical force opposing the power stroke using an isometric feedback system (Takagi et al., 2006). The initial force applied to Myo5 in this system depends on where along the actin filament Myo5 stochastically binds, allowing measurement of attachment durations at a range of resistive forces (Fig. 4 A). Plotting attachment durations as a function of force revealed a trend of longer attachment durations at higher resisting forces. At each force, attachment durations are exponentially distributed and, as expected, the data appear noisy when plotted (Fig. 4 A). Converting these data to detachment rates by binning by force, averaging, and taking the inverse clearly reveals the trend (Fig. 4 B).

The force dependence of the Myo5 detachment rate was fit by the Bell Equation:

$$k(F) = k_0 e^{\frac{-Fd}{k_B \cdot T}}, \qquad (4)$$

where $k(F)$ is the detachment rate at force $F$, $k_0$ is the detachment rate in the absence of load, $d$ is the distance parameter (the distance to the force-dependent transition state and a measure of force sensitivity), $k_B$ is Boltzmann's constant, and $T$ is the temperature. Best fit parameters for $k_0$ and $d$ were determined by MLE of the unaveraged data from Fig. 4 A, incorporating the instrument response time (Woody et al., 2016). The estimated detachment rate in the absence of force is 67.6 s$^{-1}$, in agreement with the measured detachment rate under low load

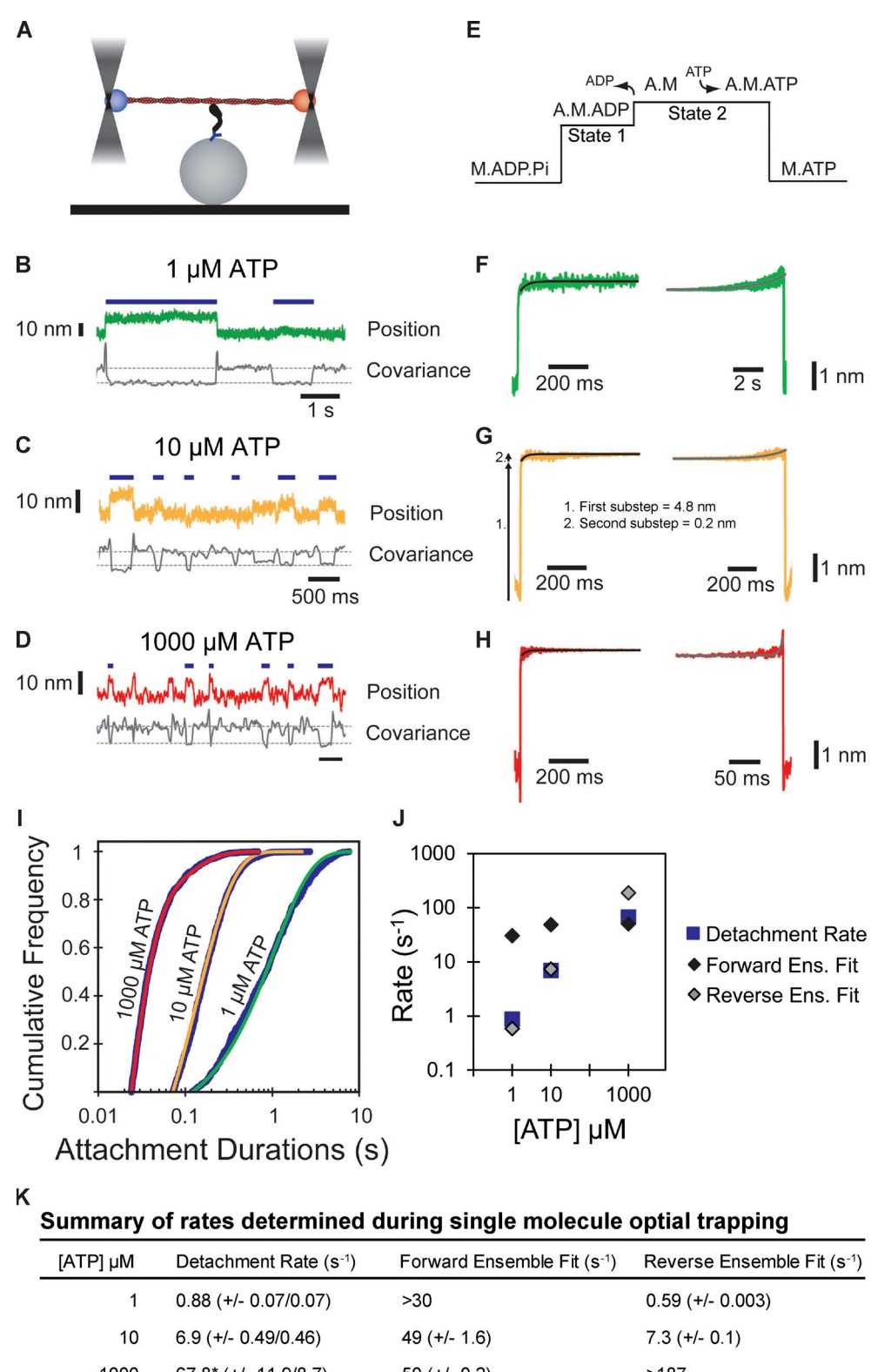

Figure 3. **Single molecule, optical trap analysis of Myo5 step size and kinetics. (A)** Cartoon schematic of the three-bead optical trapping setup. A biotinylated actin filament is tethered between two neutravidin-coated beads that are trapped in a dual-beam optical trap. This bead-actin-bead "dumbbell" is lowered onto pedestal beads that have been sparsely coated with His$_6$ antibody to attach Myo5-motor/lever-Avi-Tev-His$_9$. **(B–D)** Single Myo5 displacements of a single bead position and covariance traces, calculated using both beads, showing single molecule interactions acquired in the presence of 1 µM (B) 10 µM (C), and 1,000 µM ATP **(D)**. Blue bars indicate attachment events as identified by covariance (gray) decreases. The threshold of event detection by the covariance traces are indicated by dashed gray lines. **(E)** Schematic of displacement traces depicting the two-step nature of actomyosin displacements in the optical trap. **(F–H)** Binding events were synchronized at their beginnings (left) or ends (right) and averaged forward or backward in time, respectively. The

measured total displacement of Myo5 was 5.0 nm at 10 µM ATP, with the first substep contributing a 4.8 nm displacement (arrow 1 in G) and the second substep contributing a 0.2-nm displacement (arrow 2 in G). **(F–H)** Left: Forward-averaged ensembles synchronized at the beginnings of events. Right: Reverse-averaged ensembles synchronized at the ends of events. Black and gray lines are single exponential fits in the forward and reverse ensembles, respectively. **(I)** Cumulative distributions of attachment durations for Myo5 at 1, 10, and 1,000 µM ATP. Blue lines show the cumulative frequency of attachment durations at the indicated ATP concentrations, and the red, yellow, and green lines indicate fitted exponential distributions at 1, 10, and 1,000 µM ATP, respectively. 1 and 10 µM ATP were fit well to single exponentials, and the 1,000 µM ATP data were best described by the sum of two exponentials. **(J)** Summary of rates at 1, 10, and 1,000 µM ATP calculated from F–H. Blue boxes are the fitted exponential distributions from I, black diamonds are forward ensemble fits from F–H (left), and gray diamonds are reverse ensemble fits from F–H (right). At lower concentrations of ATP (1 and 10 µM), the rate of detachment is limited by ATP association, corresponding to the reverse ensemble fits, while at saturating ATP concentration (1,000 µM), the detachment rate is limited by the rate of ADP dissociation, corresponding to the forward ensemble fits. **(K)** Summary of rates determined via single-molecule optical trapping. Errors for detachment rates are 95% confidence intervals. Errors for forward and reverse ensemble fits are standard errors of the fits. *Detachment rates at 1,000 µM ATP were best fit to the sum of two exponents. The major component of the fit (67.8 s$^{-1}$) comprises 92.1% of the total with the remaining 7.9% having a rate of 11.6 s$^{-1}$.

conditions at 1,000 µM ATP (67.8 s$^{-1}$, Fig. 3 K), and the value for $d$ was 1.14 nm.

To put Myo5's force sensitivity in context, we replotted the function describing the force-dependent actin detachment rate of Myo5 alongside the same curves for vertebrate Myo1b, Myo1c, and β-cardiac myosin, which were determined by the same experimental approach (Fig. 4 C, Laakso et al., 2010; Greenberg et al., 2012; Woody et al., 2018). The mechanochemistry of Myo5 ($d$ = 1.14 nm) is most like that of β-cardiac (muscle) myosin

($d$ = 1.3 nm), suggesting that it is well-suited for generating power. The difference between Myo5 and acutely force-sensitive Myo1b, a tension-sensitive anchor myosin ($d$ = 15 nm), is dramatic. From 0 to 2 pN of resistance, Myo1b attachment lifetimes slow from ∼600 ms to ∼45 s, resulting in negligible power generation (Fig. 4 D). Over the same interval, Myo5 attachment lifetimes slow modestly from ∼15 to ∼25 ms, allowing it to generate considerable power (Fig. 4 D). Thus, Myo5 is unlikely to act as a force-sensitive anchor in

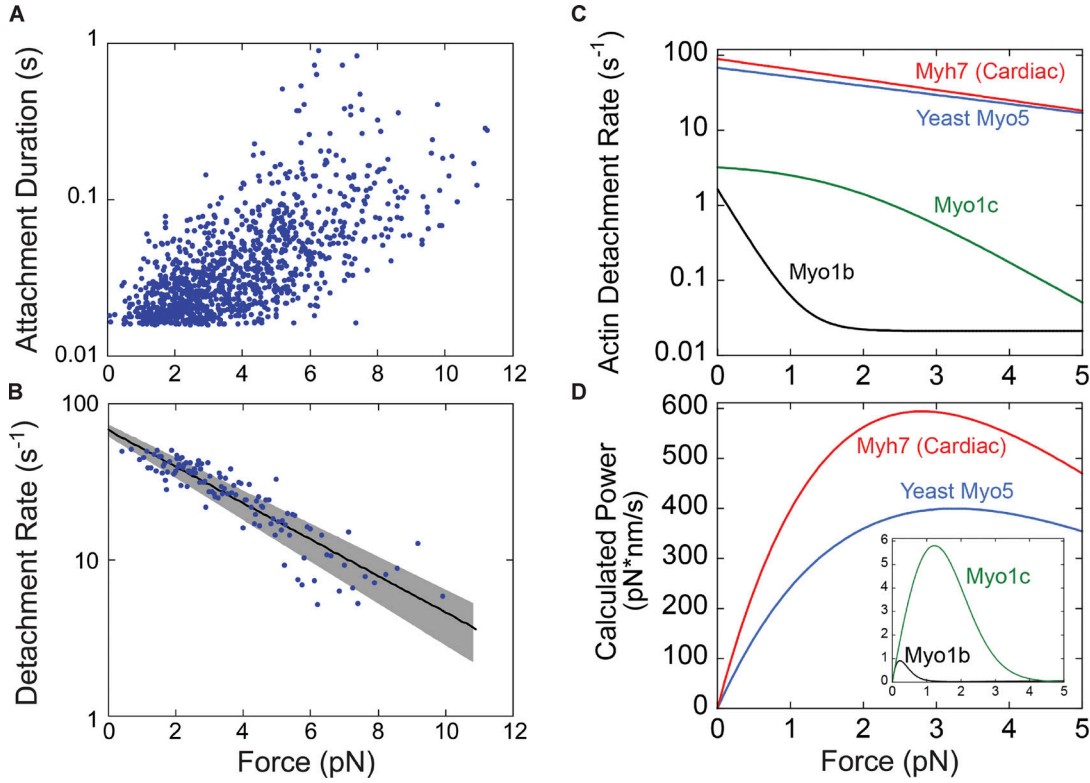

Figure 4. **Myo5 attachment lifetimes are substantially less force-dependent than other known type I myosins.** An isometric optical force clamp was utilized to determine the force sensitivity of the detachment of Myo5 from actin. **(A)** Durations of individual actomyosin attachments as a function of force, plotted on a semi-log scale. **(B)** The solid black line shows the force dependence of the detachment rates determined by MLE fitting of unaveraged points in A. For illustration purposes, attachment durations from A were binned by force at every 10 points, averaged, and converted to rates. Best-fit parameters were determined by MLE fitting and 95% confidence intervals were calculated via bootstrapping. The solid black line is calculated from best-fit parameters (k = 67.6 s$^{-1}$, d = 1.14 nm), while the gray shaded region is the 95% confidence interval (k = 62.4–72.9 s$^{-1}$, d = 1.03–1.26 nm). All MLE fitting was performed on unaveraged data and was corrected for instrument deadtime. **(C)** The force dependent detachment rate of Myo5 (from B) plotted alongside the force dependent detachment rates for Myo1b, Myo1c, and β-cardiac muscle myosin, Myh7. **(D)** Power output for the same four myosins is calculated over a range of forces by multiplying the functions from C by the applied force F, and the step size and duty ratios of each myosin.

cells and is more likely to power movements against a resisting load.

## Proposed function of type I myosin in clathrin-mediated endocytosis

Myo5 is one of the best-studied myosin-1 proteins in vivo. Extensive imaging has revealed that it is recruited to CME sites at the initiation of actin assembly, where it concentrates at the base of the site during membrane invagination (Jonsdottir and Li, 2004; Idrissi et al., 2008). Although it was known that the mechanochemical activity of type-1 myosins is required for CME (Geli and Riezman, 1996; Goodson et al., 1996; Sun et al., 2006), the mechanistic contribution of motor activity was unknown. When it was discovered that some type I myosins are acutely force-sensitive (Laakso et al., 2008), it became apparent that these motors could have mechanochemical activities that range from force-dependent anchoring to power generation. It is clear that mutant Myo5 molecules with disrupted mechanochemistry block CME, but these results do not reveal the molecular role of Myo5 (Lewellyn et al., 2015; Idrissi et al., 2012). Perhaps, the most informative finding in cells has been the observation that varying the number of type I myosins at CME sites results in altered actin assembly rates (Manenschijn et al., 2019). However, because load influences growing branched actin networks in complex ways (Bieling et al., 2016), even this finding did not clarify the molecular roles of endocytic myosin-1s.

Here, we have shown that Myo5's motor generates power rather than forming force-sensitive catch bonds. The overall ATPase rate of Myo5 is slow relative to other power-generating myosins, but its power stroke and detachment from actin are fast, and they slow only modestly under load (Fig. 4 C). Myo5's relative force insensitivity means it generates power against resistance (Fig. 4 D). Because Myo3 and Myo5 can each support CME in the absence of the other, we suspect that Myo3 is similarly force-insensitive. Given the homology between Myo5 and vertebrate Myo1e, together with the close agreement of their unloaded kinetics (El Mezgueldi et al., 2002), we also predict that Myo1e generates biologically relevant power.

Our finding that Myo5's kinetics are relatively force insensitive led us to interpret the previously described dose dependence of actin assembly on the number of myosin-1s at endocytic sites to mean that this motor moves actin filaments at CME sites to power plasma membrane invagination and create space for new monomers to assemble (Manenschijn et al., 2019; Fig. 1, right). On the order of 300 myosin molecules (Myo3 and Myo5 combined) are present at CME sites, mostly where the invaginating membrane meets the plasma membrane (Mund et al., 2018; Idrissi et al., 2008; Sun et al., 2019; Picco et al., 2015). Myo5's diffusion is likely to be impeded by the many proteins at the base of CME sites and it may move actin filaments at an angle to the membrane to which it is bound, both conditions that allow a related myosin to generate and sustain sub-piconewton forces (Pyrpassopoulos et al., 2016). Actin networks grow at the plasma membrane at ~50–100 nm/s (Kaksonen et al., 2003, 2005), so Myo5's motility rate of 700–900 nm/s (Fig. 2 G), which we would expect resistance to slow only modestly, is fast enough to do work on the actin network as it assembles. We therefore

expect that the myosins power membrane invagination and relieve the load to accelerate actin assembly during CME.

Type I myosins are involved in a variety of membrane-reshaping events in cells, where they often interact with growing branched actin networks (Sokac et al., 2006; Almeida et al., 2011; Joensuu et al., 2014; Krendel et al., 2007; Cheng et al., 2012), but the relative contributions of myosin motor activity and actin assembly have rarely been resolved. Here, we demonstrated that a type I myosin critical for CME, a process well-known to be driven by actin assembly, generates power. The implication of endocytic type I myosin as a force-insensitive motor suggests that actin assembly and myosin power generation can be coordinated to do coherent work in membrane remodeling processes.

## Materials and methods

### Reagents, proteins, and buffers

ATP concentrations were determined spectrophotometrically after each experiment by absorbance at 259 nm, $\in_{259}$ = 15,400 M$^{-1}$ cm$^{-1}$. For all ATP solutions, 1 M equivalent of MgCl$_2$ was included to make MgATP. Rabbit skeletal muscle actin was prepared and gel-filtered (Spudich and Watt, 1971). Actin concentrations were determined spectrophotometrically by absorbance at 290 nm, $\in_{290}$ = 26,600 M$^{-1}$ cm$^{-1}$. All actin was stabilized with 1 M equivalent of phalloidin (Sigma-Aldrich). Steady-state, transient, and single molecule experiments were performed at 20°C in KMg25 buffer (60 mM MOPS pH 7, 25 mM KCl, 1 mM EGTA, 1 mM MgCl$_2$, 1 mM DTT). Apyrase VII was obtained from Sigma-Aldrich. The purity and concentration of purified proteins were determined by comparing in-gel Coomassie blue staining to staining of known amounts of bovine serum albumin (Pierce).

### Expression and purification of Cmd1

The *S. cerevisiae* calmodulin gene *CMD1* was cloned from genomic DNA into a bacterial expression plasmid with a sequence encoding His$_6$-TEV situated at the 5′ end to generate pDD2743. pDD2743 was transformed into Rosetta *E. coli*, optimized for expression (Novagen). A saturated overnight culture in LB (10 g/L Bacto tryptone, 5 g/L Bacto yeast extract, 10 g/L NaCl) was used to inoculate a 1-L culture in LB to OD$_{600}$ = 0.1. Cells were grown to OD$_{600}$ = 0.6–1, induced with 0.5 mM IPTG for 5 h at 37°C, pelleted at 4,225 × *g* for 20 min at 4°C in a Sorvall SLA-3000 (fixed angle) rotor, washed with cold 20 mM HEPES pH 7.5, and re-pelleted at 2,250 × *g* for 10 min at 4°C in a Jouan CR3i (swinging bucket) centrifuge. Cell pellets were flash-frozen in 45 ml lysis buffer (20 mM HEPES pH 7.5, 1 M KCl, 20 mM Imidazole). Upon thawing, cells were lysed by sonication, 2 mg DNase I (Roche) and Triton X-100 to 1% were added, and the resulting lysate was incubated on ice for 30 min, then spun at 92,000× *g* for 25 min in a Beckman Type 70 Ti rotor. The supernatant was loaded onto a 1-ml HisTrap HP column (GE healthcare) pre-equilibrated with binding buffer (20 mM HEPES pH 7.5, 500 mM KCl, 20 mM imidazole). The column was washed with 20 ml binding buffer, and Cmd1 was eluted using a 30 ml linear gradient from 0–100% elution buffer (20 mM

HEPES pH 7.5, 500 mM KCl, 500 mM imidazole). Fractions containing Cmd1 were pooled, Cmd1 was cleaved from His$_6$ with TEV protease, and dialyzed overnight at 4°C into a low salt buffer (10 mM Tris pH 7, 25 mM NaCl, 2 mM MgCl$_2$, 5 mM DTT). Following dialysis, purified, cleaved Cmd1 was bound to a MonoQ column and eluted using a 10 ml linear gradient from 0–70% high salt buffer (10 mM Tris pH 7, 1 M NaCl, 2 mM MgCl$_2$, 5 mM DTT). Fractions containing Cmd1 were pooled, dialyzed into KMg50 buffer (60 mM MOPS pH 7, 50 mM KCl, 1 mM MgCl$_2$, 1 mM EGTA, 1 mM DTT, 5% glycerol), and stored at −80°C.

### Expression and purification of Myo5

Myo5 was coexpressed with the myosin chaperone She4 in *S. cerevisiae.* The *MYO5* open reading frame (ORF) from *S. cerevisiae* was cloned from genomic DNA and truncated at Gly[763], generating a construct containing the motor domain and both Cmd1-binding IQ motifs of the lever arm. The *SHE4* ORF was cloned in its entirety from *S. cerevisiae* genomic DNA. Both ORFs were ligated into a 2-μm expression plasmid with a partially defective *LEU2* gene (*leu2d*) to ensure a high copy number, creating plasmid pDD2744 (parent vector described in Roy et al., 2011). The *MYO5* ORF was situated with a sequence encoding AviTag-TEV-His$_9$ at the 3′ end. Expression of the *MYO5* and *SHE4* ORFs was driven by a bidirectional Gal 1/10 promotor.

pDD2744 was transformed into D1074 yeast (Roy et al., 2011). Saturated overnight cultures in synthetic minimal medium (1.5 g/L Difacto yeast nitrogen base, 5 g/L ammonium sulfate, supplemented with 2% glucose 20 μg/ml adenine, *L*-histidine, *L*-methionine, and 30 μg/ml *L*-lysine) were used to inoculate 1.5 liters of cultures in the same media with raffinose substituted for glucose to OD$_{600}$ = 0.1. After 18 h of growth at 30°C, cultures were induced with 2% galactose, Bacto yeast extract was added to 10 g/L, and Bacto peptone to 20 g/L. After 8 h of expression, the cells were harvested at 4,225 × *g* for 20 min at 4°C in a Sorvall SLA-3000 rotor, washed with 25 ml cold Milli-Q water, repelleted at 2,250 × *g* for 10 min at 4°C in a Jouan CR3i centrifuge, resuspended in 0.2 vol of cold Milli-Q water, and drop frozen into liquid nitrogen. Lysis was achieved through cryomilling (10 cycles of 3 min grinding with one min cooldown) in the large vials of a 6870 freezer/mill (SPEX Sample Prep).

Cell powders were thawed in binding buffer (10 mM Tris pH 7, 500 mM NaCl, 4 mM MgCl$_2$, 2 mM ATP, 20 mM imidazole, 5 mM DTT) supplemented with 1 mM PMSF, 1 × cOmplete protease inhibitor cocktail without EDTA (Roche), and 1 μM Cmd1. For purification of phosphorylated Myo5, 1 μg Pak1 (Sigma-Aldrich, Brzeska et al., 1997; Fig. S1) was included in the lysis buffer and 10 mM β-glycerophosphate, 5 mM sodium pyrophosphate, and 50 mM sodium fluoride were included in all purification buffers. For the purification of unphosphorylated Myo5, 4,000 units of lambda phosphatase (NEB) and 1 mM MnCl$_2$ were included in the lysis buffer. The lysate was then spun at 345,000 × *g* for 10 min at 4°C in a Beckman TLA100.3 rotor, filtered through a 0.22-μm filter, and loaded onto a 1 ml HisTrap HP column. The column was washed with wash buffer (binding buffer with only 200 mM NaCl), and Myo5 was eluted using a 20 ml linear gradient from 0-100% elution buffer (wash buffer with 1 M imidazole).

Fractions containing Myo5 were pooled and supplemented with Cmd1 to 1 μM. For unphosphorylated Myo5 purification, a further 20,000 units of lambda phosphatase were added along with MnCl$_2$ to 1 mM, and the fractions were incubated at 30°C for 30 min. Purified protein was dialyzed through a 3.5 KD MWCO membrane into 1 L storage buffer (KMg50 with 50% glycerol) overnight at 4°C and again into 500 ml of the same buffer for 2 h at 4°C and then stored at −20°C.

### Kinetic measurements

Steady-state actin-activated ATPase activity was measured using the NADH enzyme-linked assay in an Applied Photophysics SX.18 MV stopped-flow apparatus (De La Cruz and Ostap, 2009). In one reaction, the syringe contained the ATP mix (200 μM NADH, 20 U/ml lactic dehydrogenase, 100 U/ml pyruvate kinase, 500 μM phopho(enol)pyruvate, 2 mM MgCl$_2$, 2 mM ATP in KMg25) and the other syringe contained the mixture of actin (0–80 μM) and myosin (100 nM) in KMg25. The concentrations above are after mixing. After mixing, the concentration of NADH loss due to ATP hydrolysis was monitored by absorbance at 340 nm ($\in_{340}$ = 6,220 M$^{-1}$cm$^{-1}$), and the linear regions of the curve were fitted to a straight line to determine ATPase activity.

ATP-induced dissociation of actoMyo5 was measured and analyzed as described (De La Cruz and Ostap, 2009). Briefly, one reaction syringe contained ATP (0–2.7 mM) in KMg25 and the other syringe contained 100 nM Myo5 and 100 nM phalloidin-stabilized actin. Reactants were rapidly mixed by the instrument, and light scattering at 90° was acquired using a 450-nm excitation light and a 400-nm emission filter. Experimental transients were fit by single exponentials using the software provided with the stopped-flow apparatus. One to seven traces were averaged together to generate each data point. 0.04 U/ml apyrase was added to solutions of actoMyo5 before mixing to remove contaminating ADP and ATP. Unphosphorylated acto-Myo5 required prolonged treatment with apyrase to achieve sufficient signal, presumably because a larger fraction of the population was bound to ATP left over from purification and because the actin-activated ATPase rate of unphosphorylated Myo5 is slow. ADP release transients were acquired and analyzed as above by preincubating 100 μM ADP with 200 nM actoMyo5 and then rapidly mixing with 2.5 mM ATP. The concentrations reported are after mixing.

### Motility assays

Motility assays were carried out essentially as in Lin et al. (2005). Double-sided Scotch tape and vacuum grease were used to create flow chambers from a clean glass coverslip (22 mm × 40 mm, #1.5; Thermo Fisher Scientific) and a glass coverslip coated with 20 μl nitrocellulose (catalog number 11180; Ernest F. Fullam, Inc.). A mouse monoclonal antibody against His$_6$ (Sigma-Aldrich), made with 0.2 mg/ml in motility buffer (10 mM MOPS pH 7, 25 mM KCl, 1 mM EGTA, 1 mM MgCl$_2$, 1 mM DTT), was first added to the flow chamber and incubated there for 5 min to coat the nitrocellulose-coated coverslip with the antibody. The flow chamber was then blocked for 2 min with 2 mg/ml casein. Blocking coverslips with bovine serum albumin (BSA) led to inferior gliding. Phosphorylated or unphosphorylated Myo5 in

motility buffer with 2 mg/ml casein, diluted to a range of concentrations as indicated in Fig. 2 G, was added to the flow chamber and incubated for 2 min, then the chamber was washed once with motility buffer containing 1 mM ATP and 5 µM Cmd1 and three more times with the same buffer without ATP. Motility was initiated by loading the chambers with 5 nM rhodamine-phalloidin-labeled actin filaments in motility buffer with 1 µM ATP, 5 µM Cmd1, 2 mg/ml casein, 0.4 mg/ml glucose oxidase, 0.08 mg/ml catalase, and 5 mg/ml glucose. Movies of actin motility in the flow chambers were recorded at room temperature (~20°C) on a Leica DMI3000 B microscope outfitted with a 100×, 1.4 NA plan apo objective and a Retiga R6 CCD camera (Teledyne), controlled by Metamorph software. The rate of actin filament gliding was determined using the manual tracking plugin in Fiji.

## Optical trapping

Flow chambers for optical trapping were constructed with double-sided tape and vacuum grease as previously described (Snoberger et al., 2021; Greenberg et al., 2017). Briefly, the coverslip was coated with a 0.1% mixture of nitrocellulose and 2.47 µm diameter silica beads. Coverslips were dried for at least 30 min and were used within 24 h of preparation. To define the walls of the flow cell, two strips of double-sided tape were placed ~5 mm apart, and a 1-mm thick glass slide was placed on top and carefully sealed with vacuum grease after the addition of the final buffer.

Trapping buffer (KMg25 with 1 mM DTT freshly added) was used for all trapping assays. A 100× stock of glucose oxidase + catalase (GOC) was freshly prepared by centrifuging catalase (>30,000 U·ml$^{-1}$; Sigma-Aldrich) at 15,000 × $g$ for 1 min and adding 2 µl of catalase supernatant to 20 µl of 19.1 U/µl glucose oxidase (Sigma-Aldrich).

0.01 mg/ml anti-His$_6$ antibody (Sigma-Aldrich) was flowed in the chamber and incubated between 30 s and 3 min and then immediately blocked with 2, 3-min incubations of 1–2 mg/ml BSA. Stocks of phosphorylated His$_9$-tagged Myo5 were diluted to 1 nM in trapping buffer with 300 mM added KCl and incubated in the flow cell for 2 min. The number of myosins bound to the surface was limited by the surface concentration of anti-His$_6$ antibody and the incubation time of anti-His$_6$ antibody was adjusted daily between 30 s and 3 min such that one of the three to five pedestals tested showed clear myosin interactions with the actin dumbbell.

Following incubation with Myo5, a second blocking step with two 3-min incubations of 1–2 mg/ml BSA was performed. The final buffer added to the flow cell contained trapping buffer with the indicated amount of ATP, 1 µl of GOC added immediately prior to addition to the chamber, and 0.1–0.25 nM rabbit skeletal muscle actin polymerized with 15% biotinylated actin (Cytoskeleton) stabilized by rhodamine-phalloidin (Sigma-Aldrich) at a 1.1–1.2 molar ratio with G-actin concentration. Neutravidin-coated beads were prepared by incubating 0.4 ng of 0.8 µm diameter polystyrene beads (Polysciences) and coated with 5 mg/ml neutravidin (Thermo Fisher Scientific). 3 µl of neutravidin-coated beads were added to one side of the chamber prior to sealing. All trapping data were acquired within 90 min of the addition of the final buffer to the chamber.

Optical trapping experiments were performed at room temperature (20 ± 1°C) using a dual beam 1,064-nm trapping laser as previously described (Woody et al., 2017, 2018). A single laser beam was split into two beams using polarizing beam splitters and steered into a 60× water immersion objective (Nikon). Laser light was projected through an oil immersion condenser and into quadrant photodiodes (JQ-50P; Electro Optical Components, Inc.), each of which were conjugate to the back focal plane of the objective. Direct force detection from the quadrant photodiodes was achieved using a custom-built high-voltage reverse bias and an amplifier. Data acquisition, beam position control output, and isometric feedback calculations were controlled with custom-built virtual instruments (Labview; Matlab).

Individual 0.8-µm-diameter neutravidin-coated beads were caught in the two traps and held ~5 µm apart. Trap stiffnesses were adjusted to 0.05–0.1 pN/nm for each trap. A biotinylated actin filament visualized by rhodamine-phalloidin was bound to the two trapped beads, creating a bead-actin-bead dumbbell. The dumbbell was pretensioned (3–5 pN) by steering one beam using a piezo-controlled mirror conjugate to the back focal plane of the objective, and the surface of the pedestal beads was probed for myosins. Putative myosin interactions were detected via drops in the variance of the two beads, and the three-dimensional position of the dumbbell relative to the myosin was refined further by maximizing the rate and size of the observed power stroke deflections. Every 30–60 s, the dumbbell was moved axially along the actin filament in ~6-nm steps between trace acquisition to ensure even accessibility of actin-attachment target zones. Stage drift was corrected via a feedback system using a nano-positioning stage and imaging the position of the pedestal bead with nm precision (Woody et al., 2017). In experiments using 1 µM ATP, due to the longer actomyosin interactions, stage drift was still observed even with the stage feedback engaged, leading to a presumed underestimation of the displacement size. All data were acquired at a sampling rate of 250 kHz.

Isometric optical clamping experiments were performed as previously described (Woody et al., 2018; Takagi et al., 2006) using a digital feedback loop and a 1-D electro-optical deflector (EOD, LTA4-Crystal; Conoptics) to steer the beam position using input from a high-voltage source (Model 420 Amplifier; Conoptics). Briefly, the position of one bead (the "transducer" bead) was maintained at a constant position by adjusting the position of the other bead (referred to as the "motor" bead) during actomyosin interactions. The response time of the feedback loop during actomyosin interactions was ~15–30 ms.

## Optical trap data analysis

Actomyosin interactions for non-isometric optical clamping experiments were detected using the single-molecule computational tool SPASM (Software for Precise Analysis of Single Molecules, Blackwell et al., 2021), which uses a calculation of the dumbbell bead covariances and a change-point algorithm. Data collected at 1,000 µM ATP were analyzed at 250 kHz, while data collected at 1 and 10 µM ATP were downsampled to 2 kHz by averaging every 125 points to enhance analysis speed. Events were detected by calculating the covariance of the two beads

Submitted: 23 March 2023

using a smoothing window of 33.3, 15, and 5.25 ms and an averaging window 60, 36, and 12 ms at 1, 10, and 1,000 µM ATP, respectively. The instrument deadtime was calculated to be two times the covariance averaging window. For each 15-s trace, the detected covariance was plotted and fit to double Gaussian distributions, with the smaller mean Gaussian corresponding to the actomyosin "bound" portion and the larger mean Gaussian corresponding to the "unbound" portion of events. A putative event was defined as an event where the covariance starts above the unbound peak mean, drops below the bound peak mean, and remains below the unbound peak mean for at least the length of the instrument deadtime prior to returning back above the unbound peak mean. Event starts and ends were further refined using a changepoint algorithm as described (Blackwell et al., 2021). Attachment durations and ensemble averages of single events were determined using built-in features in the SPASM software. Exponential fits for forward and reverse ensemble averages were performed in Origin 2019 graphing and analysis software (OriginLab).

Events detected in isometric optical clamping experiments were detected as described in (Takagi et al., 2006) using a zero crossing analysis via custom MATLAB scripts. Briefly, when a myosin is actively engaged with the dumbbell, force is applied to the transducer bead, a feedback loop is engaged, and opposing force is applied to the motor bead until the position of the transducer bead is restored. Beginnings of events are defined at the point at which the feedback signal increases from the baseline in the motor bead and ends of events are defined when the feedback signal decreases back below the baseline in the motor bead.

### Online supplemental material

Fig. S1 shows the results of a kinase assay demonstrating that Pak1, used in purifications of phosphorylated Myo5, specifically phosphorylates Myo5 serine-357. Video 1 shows motility assays with phosphorylated Myo5. Video 2 shows motility assays with unphosphorylated Myo5.

## Acknowledgments

We thank M. Greenberg, E. Lewellyn, and A. Kunibe, each of whom played important roles at the inception of this study. We thank Y.E. Goldman for optical trap instrumentation. We thank M. Ferrin for his helpful comments on the manuscript.

This work was funded by the National Institute of General Medical Sciences (NIGMS) grant R35 GM118149 to D.G. Drubin and grant 5R37GM057247 to E.M. Ostap. R.T.A. Pedersen is currently funded by NIGMS F32 GM142145.

Author contributions: R.T. Pedersen, A. Snoberger, S. Pyrpassopoulos, D.G. Drubin, and E.M. Ostap conceived of the experiments. R.T. Pedersen, A. Snoberger, S. Pyrpassopoulos, and D. Safer generated the reagents. R.T. Pedersen, A. Snoberger, S. Pyrpassopoulos, D. Safer , and E.M. Ostap performed the experiments and analyzed the data. R.T. Pedersen, A. Snoberger, D.G. Drubin, and E.M. Ostap wrote the manuscript. D.G. Drubin and E.M. Ostap secured funding.

Disclosures: The authors declare no competing interests exist.

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

# Supplemental material

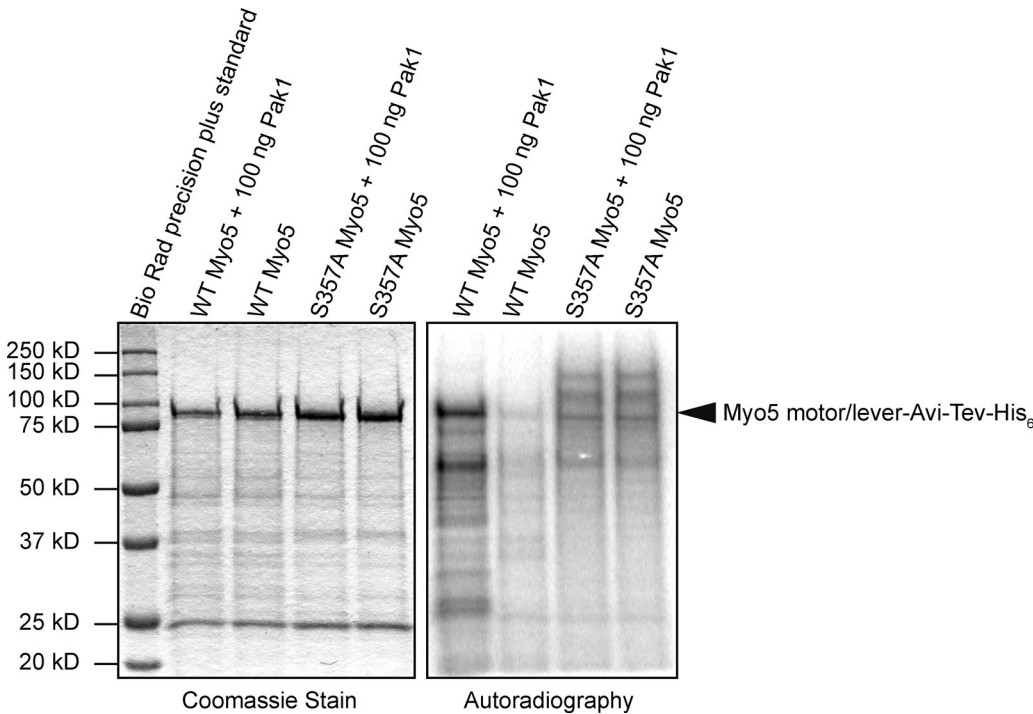

Figure S1.  **P21-activated Kinase 1 (Pak1) phosphorylates Myo5 on S357.** Crude preparations of wild type and S357A Myo5 motor/lever constructs were mixed with 250 μM ATP including 20 μCi of ATPγP32 in kinase assay buffer (5 mM MOPS pH 7, 2.5 mM β-glycerophosphate, 5 mM MgCl$_2$, 400 μM EDTA, 1 mM EGTA, and 50 μM DTT) in either the presence or absence of Pak1. Reactions were incubated at 25°C for 60 min, then quenched by adding an equal volume of 2× Tris urea sample buffer (125 mM Tris pH 6.8, 6 M urea, 2% SDS, 0.1% bromophenol blue, 10% β-mercaptoethanol) and resolved on a 10% polyacrylamide gel. The gel was stained with Coomassie, then dried onto Whatman paper and exposed to a storage phosphor screen (Amersham). The Coomassie-stained gel was imaged on a standard photo scanner and the phosphor screen on a Typhoon gel imager (Amersham). Note that there are differences in baseline labeling in the absence of added kinase between the two different protein preps, but the addition of Pak1 clearly results in radiolabeling of wild type but not mutant Myo5.

Video 1.   **Motility assays with phosphorylated Myo5.** Rhodamine-phalloidin-labeled actin filaments gliding over coverslips coated with a concentration series of phosphorylated Myo5 protein in motility buffer with 1 mM ATP. Videos were collected at one frame per second and are played back at 16 frames per second.

Video 2.   **Motility assays with unphosphorylated Myo5.** Rhodamine-phalloidin-labeled actin filaments gliding over coverslips coated with a concentration series of unphosphorylated Myo5 protein in motility buffer with 1 mM ATP. Short movies of motility at 100 and 150 nM unphosphorylated Myo5 were collected because no motility was observed. Movies at all other concentrations were collected at one frame every 4 s and are played back at 16 frames per second. The playback rate of Video 2 is four times faster than the playback rate of Video 1.

