## [Peer Review File · The Journal of Cell Biology]

Endocytic myosin-1 is a force-insensitive, power-generating motor

Ross Pedersen, Aaron Snoberger, Serapion Pyrpasopoulos, Daniel Safer, David Drubin, and E Michael Ostap

Corresponding Author(s): E Michael Ostap, University of Pennsylvania and David Drubin, University of California, Berkeley

Review Timeline:

Submission Date:	2023-03-23
Editorial Decision:	2023-04-25
Revision Received:	2023-05-17
Editorial Decision:	2023-06-21
Revision Received:	2023-06-30

Monitoring Editor: Alex Mogilner

Scientific Editor: Tim Fessenden

Transaction Report:

DOI: <https://doi.org/10.1083/jcb.202303095>

April 25, 2023

Re: JCB manuscript #202303095

Dr. E Michael Ostap
University of Pennsylvania
Pennsylvania Muscle Institute, Perelman School of Medicine
740 Clinical Research Building
415 Curie Boulevard
Philadelphia, Pennsylvania 19104

Dear Dr. Ostap,

Thank you for submitting your manuscript entitled "An endocytic myosin essential for plasma membrane invagination powers motility against resistance". The manuscript was assessed by expert reviewers, whose comments are appended to this letter. We invite you to submit a revision if you can address the reviewers' key concerns, as outlined here.

As you will see, reviewers were generally pleased with the fundamental advance into the properties of Myo5. However, Reviewers 1 and 2 felt additional evidence was needed to link these measurements with the role for this motor in CME, and we agree that a revision must include additional observations to fill this gap. However, additional single-molecule force measurements are not required in a revision. In addition, reviewers offered suggestions concerning the phosphorylation status of this myosin, which we agree are intriguing but are not required.

GENERAL GUIDELINES:

Text limits: Character count for a Report is < 20,000, not including spaces. Count includes title page, abstract, introduction, the joint Results & Discussion, and acknowledgments. Count does not include materials and methods, figure legends, references, tables, or supplemental legends.

Figures: Reports may have up to 5 main text figures. To avoid delays in production, figures must be prepared according to the policies outlined in our Instructions to Authors, under Data Presentation, <https://jcb.rupress.org/site/misc/ifora.xhtml>. All figures in accepted manuscripts will be screened prior to publication.

Supplemental information: There are strict limits on the allowable amount of supplemental data. Reports may have up to 3 supplemental figures. Up to 10 supplemental videos or flash animations are allowed. A summary of all supplemental material should appear at the end of the Materials and methods section.

Please note that JCB now requires authors to submit Source Data used to generate figures containing gels and Western blots with all revised manuscripts. This Source Data consists of fully uncropped and unprocessed images for each gel/blot displayed in the main and supplemental figures. Since your paper includes cropped gel and/or blot images, please be sure to provide one Source Data file for each figure that contains gels and/or blots along with your revised manuscript files. File names for Source Data figures should be alphanumeric without any spaces or special characters (i.e., SourceDataF#, where F# refers to the associated main figure number or SourceDataFS# for those associated with Supplementary figures). The lanes of the gels/blots should be labeled as they are in the associated figure, the place where cropping was applied should be marked (with a box), and molecular weight/size standards should be labeled wherever possible. Source Data files will be made available to reviewers during evaluation of revised manuscripts and, if your paper is eventually published in JCB, the files will be directly linked to specific figures in the published article.

The typical timeframe for revisions is three to four months. While most universities and institutes have reopened labs and allowed researchers to begin working at nearly pre-pandemic levels, we at JCB realize that the lingering effects of the COVID-19 pandemic may still be impacting some aspects of your work, including the acquisition of equipment and reagents. Therefore,

if you anticipate any difficulties in meeting this aforementioned revision time limit, please contact us and we can work with you to find an appropriate time frame for resubmission. Please note that papers are generally considered through only one revision cycle, so any revised manuscript will likely be either accepted or rejected.

Thank you for this interesting contribution to Journal of Cell Biology. You can contact us at the journal office with any questions, cellbio@rockefeller.edu or call (212) 327-8588.

Sincerely,

Alex Mogilner
Monitoring Editor
Journal of Cell Biology

Tim Fessenden
Scientific Editor
Journal of Cell Biology

Reviewer #1 (Comments to the Authors (Required)):

Class I myosins (Myo1s) are critical for organizing the actin network around a forming clathrin coated pit and powering its invagination, a process that has been well-studied in budding yeast. In spite of a significant amount of work to understand the mechanism of endocytosis, the precise role of Myo1 in endocytic membrane invagination remains unclear. Two competing models for how Myo1 acts propose that it either as a power generator, sliding actin filaments against the membrane to facilitate actin polymerization by pulling them back from the plasma membrane or as a force-sensitive anchor that strengthens the linkage to actin upon sensing force, helping to maintain organization/orientation of actin filaments associated with the invaginating vesicle. Distinguishing between these two models absolutely requires detailed characterization of the kinetic and biophysical properties of motor.

The paper by Pedersen et al describes a careful, detailed study of the kinetic and biophysical properties of budding yeast Myo5 that has a role in clathrin mediated endocytosis (it is functionally redundant with the closely related Myo3). The results reveal that, as would be expected for a Myo1, Myo5 is low duty ratio motor (i.e. it spends a small fraction of its enzymatic cycle on actin) and has a two-step working stroke. Optical trapping experiments show that the rate of the first 4.8 nm step corresponds to that of ADP release with the lifetime of the second, smaller 0.2 nm step corresponding to that of ATP binding. An isometric feedback system was used to measure Myo5 attachment with increasing applied force. This technically demanding experiment, that the Ostap lab is well-experienced in performing, shows that Myo5 is highly insensitive to applied F. All together these finding clearly establish that Myo5p is a power generating motor. Given the high degree of evolutionary conservation between Myo5 and mammalian Myo1E, the findings suggest that Myo1E operates in a similar manner.

The lack of force sensitivity by Myo5 is consistent with the model of the motor acting to pull actin filaments away from plasma membrane enabling monomer addition at the plasma membrane to promote continued actin filament growth that drives vesicle invagination. However, the results do not add to our understanding of exactly how Myo5, a low duty monomeric motor, can do that. The paper does not provide any direct test of the likely model for Myo5 action. The conclusions would be strengthened by some complementary in vivo work, even just a limited experiment that might have arisen from the findings here to support the conclusions.

It should be noted, however, that the force measurements are technically challenging and the response to force has only been tested for a handful of Myo1s. The kinetic parameters and step size are also a critical characteristics that one needs to know for fully understanding how a motor can act in a cell. In other words, this type of work does need to be done and the authors effectively put their work in the full cell biological context.

Additional comments -

It would have been helpful if the authors could have gone a bit further in explaining (speculating) how a single motor that quickly

detaches from actin could meaningfully generate sufficient F to pull an actin filament back from the membrane when that filament would likely be embedded in a network that would possibly provide significant resistance. Or if they posit that a small collective of motors working together would be necessary and, if so, what would bring them together.

Methods - centrifugation steps should describe the x g centrifugal force and not simply list an instrument and/or rotor.

Reviewer #2 (Comments to the Authors (Required)):

Review for Pedersen et al. "An endocytic myosin essential for plasma membrane invagination powers motility against resistance" for Journal of Cell Biology.

In this short manuscript, the authors are reporting in vitro experiments to decorticate whereas Myo5, a cerevisiae equivalent of mammalian myosin 1, is a force-sensitive anchor or a power generator.

The experiments in vitro are well done and clearly presented. The results are convincing up to figure 4. The laser tweezers experiments in particular are very nice and show clearly heavy dependence on ATP concentration for the detachment rate of Myo5.

The main message that the author want to bring forward is then coming in the last figure 4.

This is the first major weak point. This unique experiment is both conclusive (it shows a dependence of detachment rate in function of force) and non-conclusive as the author say the opposite by comparing their results with existing knowledge on other myosin. We have no comparison with other myosins done with the same experimental setup to see sensitivity of the system, nor mutants Myo5 (rigor mutant for example). We have also no alternative approaches to strengthen this point as it is the main point of the manuscript and it would be welcome here.

The second major weak point is the overall writing that is misleading. Title/abstract catch your attention and advertise a paper talking of endocytosis, membrane invagination, motility against resistance. But the real content of the paper is "Myo5 from yeast is a rapid motor with weak sensitive anchorage". The rest of the message is not supported by the experiments but by the existing literature to try to integrate the new results. It is not fine for me to carry such a large and vague message to end up presenting few in vitro, protein-only based experiments (don't get me wrong here, nothing wrong here of having only in vitro experiment in a scientific report, the wrongdoing is the text not the experiments). The paper title and abstract have to be edited completely to fit with the narrow and very specific findings presented in the paper. The interpretation and integration of the results are part of the discussion only, or the author can write latter a review integrating there finding if they want to have such large catchy message, but it's not acceptable as presented.

Concerning the integration of the results in the mechanism of endocytosis, I may have a simpler understanding of the process and of the advantage of having a rapid motor than what is proposed. It is hard for me to see Myo5 as capable to displacing actin filaments in the in vivo context. Whatever the force this myosin is capable, those myosin works as monomer and in order to apply torque, the myosin need to be anchor to something that resist. Myo5 is most likely kinked to the membrane on the other side and given the fluid aspect of the bilayer, this torque will only work if you are normal to the membrane plane and that the membrane resist bending. On the contrary, having a fast myosin (from in vitro it is capable to move filaments at 500 to 1000 nm/s in the experiment presented) is very advantageous to shape the membrane tightly around actin and to keep polymerization at bay in the "cul de sac" that represent the transition zone between the invaginated membrane and the plasma membrane. This zone is also geometrically reminiscent of the lamellipodia tip where myosin1e in mammals is known to locate. The max speed of actin polymerization in mammals is around 150-170 nm/s for arp2/3 driven processes. Having a myosin1 that cope easily with this actin speed to keep membrane tight around actin at the tip of polymerization will be advantageous to keep polymerization focused at the right place and avoid dispersion of the energy and force needed to promote and finish endocytosis. The myosins working collectively, bound to the membrane, will constantly move toward the barbed end tips, forcing the membrane to stay tight along actin, explaining easily why in vivo Myo5 is located mostly at the base of the endocytic pit at the transition zone. The plasma membrane will serve as the "buffer zone" where many myosins can be recruited by stochastically binding to the membrane and, once encountering actin, will move toward the tips keeping all that organization in control, otherwise everything could go more randomly and you could end-up with uncontrolled and ineffective actin polymerization at the wrong place like what was described for phagocytosis in myo1e/f KO macrophages (Barger et al., 2020, Nat Comm).

Reviewer #3 (Comments to the Authors (Required)):

The manuscript by Pedersen et al characterizes yeast Myo5 using a variety of biochemical and biophysical methods. The study is timely and well done.

There are two main points of the paper. The first is that phosphorylation is required for full activity (strong support, but a few additional experiments would make an even stronger case).

The powerstroke is relatively force insensitive which is important in defining the possible roles of this myosin in cells (strongly

supported).

I have only a few questions and comments.

How was the extent of phosphorylation determined?

Fig. 2g. Another interpretation of this could be that there is a small percentage of phosphorylated (or otherwise "active" Myo5 in the prep that is giving rise to the slower motility and that the reason for the reduced velocity is that this small amount of active myosin is working against a frictional nonproductive binding of the inactive molecules in the prep. In this regard it might be interesting to examine the motility of a mixture of phosphorylated and unphosphorylated Myo5 at 100 nM each applied to the surface to see how the unphosphorylated protein affects the ensemble gliding ability.

Did you try to optically trap the unphosphorylated Myo5? I'd be curious to see whether you detect longer lived events or just saw very short transient "bumps" or no detectable interactions.

Reviewer #1 (Comments to the Authors (Required)):

Class I myosins (Myo1s) are critical for organizing the actin network around a forming clathrin coated pit and powering its invagination, a process that has been well-studied in budding yeast. In spite of a significant amount of work to understand the mechanism of endocytosis, the precise role of Myo1 in endocytic membrane invagination remains unclear.

Two competing models for how Myo1 acts propose that it either as a power generator, sliding actin filaments against the membrane to facilitate actin polymerization by pulling them back from the plasma membrane or as a force-sensitive anchor that strengthens the linkage to actin upon sensing force, helping to maintain organization/orientation of actin filaments associated with the invaginating vesicle. Distinguishing between these two models absolutely requires detailed characterization of the kinetic and biophysical properties of motor.

The paper by Pedersen et al describes a careful, detailed study of the kinetic and biophysical properties of budding yeast Myo5 that has a role in clathrin mediated endocytosis (it is functionally redundant with the closely related Myo3). The results reveal that, as would be expected for a Myo1, Myo5 is low duty ratio motor (i.e. it spends a small fraction of its enzymatic cycle on actin) and has a two-step working stroke. Optical trapping experiments show that the rate of the first 4.8 nm step corresponds to that of ADP release with the lifetime of the second, smaller 0.2 nm step corresponding to that of ATP binding. An isometric feedback system was used to measure Myo5 attachment with increasing applied force. This technically demanding experiment, that the Ostap lab is well-experienced in performing, shows that Myo5 is highly insensitive to applied F.

All together these finding clearly establish that Myo5p is a power generating motor. Given the high degree of evolutionary conservation between Myo5 and mammalian Myo1E, the findings suggest that Myo1E operates in a similar manner.

The lack of force sensitivity by Myo5 is consistent with the model of the motor acting to pull actin filaments away from plasma membrane enabling monomer addition at the plasma membrane to promote continued actin filament growth that drives vesicle invagination. However, the results do not add to our understanding of exactly how Myo5, a low duty monomeric motor, can do that. The paper does not provide any direct test of the likely model for Myo5 action. The conclusions would be strengthened by some complementary in vivo work, even just a limited experiment that might have arisen from the findings here to support the conclusions.

It should be noted, however, that the force measurements are technically challenging and the response to force has only been tested for a handful of Myo1s. The kinetic parameters and step size are also a critical characteristics that one needs to know for fully understanding how a motor can act in a cell. In other words, this type of work does need to be done and the authors effectively put their work in the full cell biological context.

This reviewer succinctly and accurately describes the main findings of our study. We thank this individual for the generous assessment of our manuscript.

We agree that the results of our study are best interpreted in the context of *in vivo* experiments. We have tried to consider how additional *in vivo* data would fit into this paper. However, Myo5 is already one of the most thoroughly studied myosin-I proteins, having been subjected to scores of detailed, quantitative *in vivo* studies that explore its molecular role in endocytosis. These studies include: (a) determination of the time course of Myo5 recruitment during endocytosis in relation to other endocytic components and membrane morphology, (b) quantification of myosin number at the endocytic site, (c) Myo5 knockout experiments that assess endocytic form and function, (d) Myo5 mutation studies that alter motor activity, and (e) manipulation of motor concentration to affect endocytic time courses. Our biophysical characterization now provides the mechanochemical details that will help the field interpret this very substantial corpus of published *in vivo* results.

Nevertheless, the comments of the Reviewer make it clear that we did not do an adequate job explaining how our work adds to *in vivo* studies. Therefore, we have added additional discussion and interpretation of the existing literature to our manuscript (lines 291-309 and 319-323). We explicitly discuss how it was determined that myosin-I motor activity is required for clathrin-mediated endocytosis and how yeast strains relying solely on mutant versions of the *MYO5* gene with amino acid substitutions or complete deletions of the motor domain have been generated and found to be completely deficient in vesicle internalization. We also highlight how the number of active motors present at endocytic sites is known and has been systematically varied by altering the copy number of the two budding yeast genes that encode type I myosins, *MYO3* and *MYO5*, revealing a roughly linear relationship between the speed of membrane invagination and the number of myosin-I molecules at endocytic sites (Manenschijn et al., *eLife*, 2019).

Given the wealth of *in vivo* analysis of the endocytic functions of these myosins in yeast we believe that the highest priority now is analysis of the biophysical properties of this protein, which then make possible interpretation of the *in vivo* data. Since type I myosin functions in endocytosis from yeast to humans, as well as in a host of other cellular processes, what is learned from these studies will be broadly impactful.

Additional comments -

It would have been helpful if the authors could have gone a bit further in explaining (speculating) how a single motor that quickly detaches from actin could meaningfully generate sufficient F to pull an actin filament back from the membrane when that filament would likely be embedded in a network that would possibly provide significant resistance. Or if they posit that a small collective of motors working together would be necessary and, if so, what would bring them together.

In vivo experiments show convincingly that Myo5 molecules at an endocytic site work in teams of nearly 100 motors. In fact, when one considers the presence of Myo3, motor numbers reach the 100s (Picco et al., *eLife* 2015, Manenschin et al., *eLife*, 2019, Sun et al., *eLife*, 2019). Given that the motors are recruited to this region, and that their diffusion in the membrane is likely limited by adaptors and protein packing, it is highly likely that the powerful motors work to displace the Arp2/3 complex-derived branched actin networks. We added a section near the end of the Results and Discussion discussing this important point (page 17).

Methods - centrifugation steps should describe the x g centrifugal force and not simply list an instrument and/or rotor.

Done.

Reviewer #2 (Comments to the Authors (Required)):

Review for Pedersen et al. "An endocytic myosin essential for plasma membrane invagination powers motility against resistance" for Journal of Cell Biology.

In this short manuscript, the authors are reporting in vitro experiments to decorticate whereas Myo5, a cerevisiae equivalent of mammalian myosin 1, is a force-sensitive anchor or a power generator.

The experiments in vitro are well done and clearly presented. The results are convincing up to figure 4. The laser tweezers experiments in particular are very nice and show clearly heavy dependence on ATP concentration for the detachment rate of Myo5.

We would like to thank this reviewer for the overall positive assessment of our manuscript.

The main message that the author want to bring forward is then coming in the last figure 4. This is the first major weak point. This unique experiment is both conclusive (it shows a dependence of detachment rate in function of force) and non-conclusive as the author say the opposite by comparing their results with existing knowledge on other myosin. We have no comparison with other myosins done with the same experimental setup to see sensitivity of the system, nor mutants Myo5 (rigor mutant for example). We have also no alternative approaches to strengthen this point as it is the main point of the manuscript and it would be welcome here.

We apologize for the confusion. Our revised text more thoroughly compares the force sensitivity of Myo5 to other myosins and discusses in greater depth the implications of Myo5 mechanochemistry on its in vivo function (see response to Reviewer 1). Importantly, other myosin paralogs have been studied with the same experimental approach to assess the force-dependence of actin detachment (e.g., Veigel et al., *Nat. Cell Biol.*, 2003; Laakso et al., *Science*, 2008 and *PNAS*, 2010; Greenberg et al., *PNAS*, 2012; Greenberg et al., *Biophys. J.*, 2014). We measured the force-dependence of the actin detachment rate because it directly impacts motility speeds, the duty ratio, and thus the amount of power a myosin can generate. Our experiments allow us to conclude that the mechanochemistry of Myo5 is similar to the powerful myosin that contracts the human heart (Myh7) but is very dissimilar to the endosome-localized, vertebrate Myo1b which acts as a force dependent anchor. Thus, our work suggests that Myo5 motors can generate power under several pN of load. If Myo5 was like vertebrate Myo1b, it would "lock" onto actin filaments and stay attached for *nearly a minute* under < 2 pN of load, acting as a force-dependent anchor rather than a motor for moving actin networks.

To illustrate our important conclusions, we added two figure panels (Figures 4C and 4D) which plot the detachment rate and calculated power output of Myo5 and other myosins the Ostap Lab has characterized previously. These plots clearly compare the force-dependent activity of Myo5 to force-dependent activities of other myosins.

Our expectation is that the quantitative in vivo measurements made by the Drubin, Kaksonen, and other labs, and mechanochemical parameters determined here by the Ostap Lab, will be used by computational biologists to fully model how the actin cytoskeleton participates in endocytosis, a classic question in cell biology.

The second major weak point is the overall writing that is misleading. Title/abstract catch your attention and advertise a paper talking of endocytosis, membrane invagination, motility against resistance. But the real content of the paper is "Myo5 from yeast is a rapid motor with weak sensitive anchorage". The rest of the message is not supported by the experiments but by the existing literature to try to integrate the new results. It is not fine for me to carry such a large and vague message to end up presenting few in vitro, protein-only based experiments (don't get me wrong here, nothing wrong here of having only in vitro experiment in a scientific report, the wrongdoing is the text not the experiments). The paper title and abstract have to be edited completely to fit with the narrow and very specific findings presented in the paper. The interpretation and integration of the results are part of the discussion only, or the author can write latter a review integrating there finding if they want to have such large catchy message, but it's not acceptable as presented.

A substantial body of work published over more than a decade has clearly shown that Myo5 is essential for plasma membrane invagination during yeast endocytosis, so the statement in the title is not new or controversial. The title also says that this motor is able to power motility under load, which is supported by the data presented in the paper. Nevertheless, as requested by this Reviewer, we have altered the Title and Abstract to make it clearer that the manuscript presents an in vitro study. Additionally, in this revision, we tried to better illustrate the implications of biophysical results via the inclusion of new panels in Figure 4 (see response to previous comment). We also edited the introduction and moved some of the more speculative points related to of biological implications to the Results and Discussion section. We hope our conclusions are now clearer to the Reviewer, and that they do not feel misled by the title or abstract.

Finally, we note that our manuscript follows the Report format, so it has a combined Results and Discussion section. Accordingly, we include interpretation and biological context for our results as they are presented in addition to several paragraphs of dedicated discussion at the end of the section.

Concerning the integration of the results in the mechanism of endocytosis, I may have a simpler understanding of the process and of the advantage of having a rapid motor than what is proposed. It is hard for me to see Myo5 as capable to displacing actin filaments in the in vivo context. Whatever the force this myosin is capable, those myosin works as monomer and in order to apply torque, the myosin need to be anchor to something that resist. Myo5 is most likely kinked to the membrane on the other side and given the fluid aspect of the bilayer, this torque will only work if you are normal to the membrane plane and that the membrane resist bending. On the contrary, having a fast myosin (from in vitro it is capable to move filaments at 500 to 1000 nm/s in the experiment presented) is very advantageous to shape the membrane tightly around actin and to keep polymerization at bay in the "cul de sac" that represent the transition zone between the invaginated membrane and the plasma membrane. This zone is also geometrically reminiscent of the lamellipodia tip where myosin Ie in mammals is known to locate. The max speed of actin polymerization in mammals is around 150-170 nm/s for arp2/3 driven processes. Having a myosin I that cope easily with this actin speed to keep membrane tight around actin at the tip of polymerization will be advantageous to keep polymerization focused at the right place and avoid dispersion of the energy and force needed to promote and finish endocytosis. The

myosins working collectively, bound to the membrane, will constantly move toward the barbed end tips, forcing the membrane to stay tight along actin, explaining easily why in vivo Myo5 is located mostly at the base of the endocytic pit at the transition zone. The plasma membrane will serve as the "buffer zone" where many myosins can be recruited by stochastically binding to the membrane and, once encountering actin, will move toward the tips keeping all that organization in control, otherwise everything could go more randomly and you could end-up with uncontrolled and ineffective actin polymerization at the wrong place like what was described for phagocytosis in myo1e/f KO macrophages (Barger et al., 2020, Nat Comm).

Thank you for this detailed speculation. This is in fact the kind of thoughtful interpretation that we hoped to foster by presenting our in vitro results in the context of an in vivo biological problem. Our intention was not to show that Myo5 works as a single motor. Rather, substantial in vivo evidence suggests that greater than 100 Myo5 motors work together in a single endocytic pit to power membrane invagination (see above), and that altering their numbers impacts endocytosis (Manenschin et al., *eLife*, 2019). Although the motors do not form filaments, one way to think of their function might be analogous to the 300 myosins in a single muscle sarcomere, wherein each motor likewise has a low duty ratio, but together they support powerful sarcomeric contractions.

We also note here that Pedersen and Drubin (*JCB*, 2019) reported disorganized actin assembly and membrane-free cytoplasmic actin comets originating from endocytic sites in budding yeast cells lacking both type I myosins or relying on a type I myosin with the membrane binding domain deleted, but mutants relying on a type I myosin with the motor domain deleted did not have the disorganized actin phenotype. This observation led us to conclude that Myo5 plays a motor independent role in organizing and corralling actin assembly at endocytic sites. We also note that in addition to having a membrane-binding domain, this myosin is a multivalent protein with multiple protein-binding domains in addition to the membrane-binding domain and therefore is likely to be integrated into a complex protein network.

Finally, we want to stress that we endeavored to cast our results in rich cell biological context specifically to invite interpretation and speculation and to inspire further mechanistic studies, including the kinds of experiments that might reveal that Myo5 is made more processive through multiple motors functioning in coordination.

Reviewer #3 (Comments to the Authors (Required)):

The manuscript by Pedersen et al characterizes yeast Myo5 using a variety of biochemical and biophysical methods. The study is timely and well done.

There are two main points of the paper. The first is that phosphorylation is required for full activity (strong support, but a few additional experiments would make an even stronger case). The power stroke is relatively force insensitive which is important in defining the possible roles of this myosin in cells (strongly supported).

Thank you for the kind words concerning our manuscript. We agree with the Reviewer that these are two important findings of this paper. We believe these new insights are critical to interpreting existing studies of this motor and understanding its role in the biological process it carries out. The effect of phosphorylation on Myo5 activity is an important aspect of the manuscript; however, we think working out the detailed mechanism of phosphorylation-dependent activation is beyond the scope of the paper. It is highly likely that the mechanism of activation is the same as for *Acanthamoeba* myosin-IC (Ostap et al., 2002), which shows very similar steady-state kinetic changes upon phosphorylation. Detailed biochemical studies of *Acanthamoeba* myosin-I demonstrate that phosphorylation directly activates the phosphate release step. We agree that these are important details, but substantial kinetic and single-molecule experiments will need to be performed.

I have only a few questions and comments.

How was the extent of phosphorylation determined?

We thank the Reviewer for this question, as this point was not sufficiently described in the text.

Our initial ATP-induced actin dissociation experiments revealed light scattering transients that were best fit by double exponential functions that differed in rate by nearly 3-fold. Treatment with a phosphatase eliminated the fast phase of the double exponential. We realized that our initial preps of Myo5 were a mixed population of phosphorylated and unphosphorylated enzymes (see Table 1 for rate constants for phosphorylated and unphosphorylated enzymes).

We suspected that the relevant phosphorylation site was the so-called TEDS rule site first described by Bement and Mooseker (*Cell Motil. Cytoskel.*, 1995), which, for other type I myosins, can be phosphorylated by p21 activated kinases (PAKs, Brzeska et al., *PNAS*, 1997). Using kinase assays with ATP-gamma-P32, we found that the Pak1 we purchased from Sigma specifically phosphorylated the TEDS rule site on Myo5, as determined via comparison with an alanine mutation at the TEDS site (S357A). We have now added data from these kinase assays as a supplemental figure.

The extent of phosphorylation was evaluated by assessing the ATP-induced actomyosin dissociation transients. Phosphatase-treated proteins (100% dephos) showed single-exponential transients described by the rate constant $K_1k_2' = 1.1 \mu\text{M/s}$, and PAK-treated proteins (100% phos) showed single-exponential transients described by the rate constant $K_1k_2' = 0.39 \mu\text{M/s}$.

Fig. 2g. Another interpretation of this could be that there is a small percentage of phosphorylated

(or otherwise "active" Myo5 in the prep that is giving rise to the slower motility and that the reason for the reduced velocity is that this small amount of active myosin is working against a frictional nonproductive binding of the inactive molecules in the prep. In this regard it might be interesting to examine the motility of a mixture of phosphorylated and unphosphorylated Myo5 at 100 nM each applied to the surface to see how the unphosphorylated protein affects the ensemble gliding ability.

This is an interesting possibility. However, given that we are confident that our proteins are nearly 100% dephosphorylated or 100% phosphorylated, it is unlikely. Additionally, we predict that the duty ratio of the dephosphorylated myosin is 10-fold lower than that of the phosphorylated myosin, so it is unlikely that the proteins are putting substantial load on gliding filaments. Nevertheless, we now mention this possibility in the manuscript text (lines 185-186).

Did you try to optically trap the unphosphorylated Myo5? I'd be curious to see whether you detect longer lived events or just saw very short transient "bumps" or no detectable interactions.

Preliminary experiments with unphosphorylated proteins did not reveal interactions, presumably because of the low probability of binding due to the low duty ratio. However, future experiments will hopefully allow us to determine if the slow gliding motility is the result of prolonged time of attachments or the slow attachment rates.

June 21, 2023

RE: JCB Manuscript #202303095R

Dr. E Michael Ostap
University of Pennsylvania
Pennsylvania Muscle Institute, Perelman School of Medicine
740 Clinical Research Building
415 Curie Boulevard
Philadelphia, Pennsylvania 19104

Dear Dr. Ostap:

Thank you for submitting your revised manuscript entitled "Power generation by a myosin-1 essential for endocytosis: implications for biological mechanism". We would be happy to publish your paper in JCB pending final revisions necessary to meet our formatting guidelines and addressing a final comment by Reviewer 1 (see details below).

A. MANUSCRIPT ORGANIZATION AND FORMATTING:

Full guidelines are available on our Instructions for Authors page, <http://jcb.rupress.org/submission-guidelines#revised>. Submission of a paper that does not conform to JCB guidelines will delay the acceptance of your manuscript.

- 1) Text limits: Character count for Reports is < 20,000, not including spaces. Count includes abstract, introduction, results, discussion, and acknowledgments. Count does not include title page, figure legends, materials and methods, references, tables, or supplemental legends.
- 2) Figures limits: Reports may have up to five main figures and three supplemental figures/tables.
- 3) Figure formatting: Scale bars must be present on all microscopy images, including inset magnifications. Molecular weight or nucleic acid size markers must be included on all gel electrophoresis.
- 4) Statistical analysis: Error bars on graphic representations of numerical data must be clearly described in the figure legend. The number of independent data points (n) represented in a graph must be indicated in the legend. Statistical methods should be explained in full in the materials and methods. For figures presenting pooled data the statistical measure should be defined in the figure legends. Please also be sure to indicate the statistical tests used in each of your experiments (either in the figure legend itself or in a separate methods section) as well as the parameters of the test (for example, if you ran a t-test, please indicate if it was one- or two-sided, etc.). Also, if you used parametric tests, please indicate if the data distribution was tested for normality (and if so, how). If not, you must state something to the effect that "Data distribution was assumed to be normal but this was not formally tested."
- 5) Abstract and title: The abstract should be no longer than 160 words and should communicate the significance of the paper for a general audience. The title should be less than 100 characters including spaces. Make the title concise but accessible to a general readership.
** The present title does not convey the findings in the paper. We suggest:
"Myosin-1 is a force-insensitive, power-generating motor."
- 6) Materials and methods: Should be comprehensive and not simply reference a previous publication for details on how an experiment was performed. Please provide full descriptions in the text for readers who may not have access to referenced manuscripts.
** Please describe in full the NADH enzyme-linked assay procedure and analysis (under Kinetic measurements), Motility assays, Optical trapping, and Optical trap data analysis.
- 7) Please be sure to provide the sequences for all of your primers/oligos and RNAi constructs in the materials and methods. You must also indicate in the methods the source, species, and catalog numbers (where appropriate) for all of your antibodies. Please also indicate the acquisition and quantification methods for immunoblotting/western blots.
- 8) Microscope image acquisition: The following information must be provided about the acquisition and processing of images:
 - a. Make and model of microscope
 - b. Type, magnification, and numerical aperture of the objective lenses

- c. Temperature
- d. Imaging medium
- e. Fluorochromes
- f. Camera make and model
- g. Acquisition software
- h. Any software used for image processing subsequent to data acquisition. Please include details and types of operations involved (e.g., type of deconvolution, 3D reconstitutions, surface or volume rendering, gamma adjustments, etc.).

10) Supplemental materials: There are strict limits on the allowable amount of supplemental data. Reports may have up to 3 supplemental figures. Please also note that tables, like figures, should be provided as individual, editable files. A summary of all supplemental material should appear at the end of the Materials and methods section.

13) ORCID IDs: ORCID IDs are unique identifiers allowing researchers to create a record of their various scholarly contributions in a single place. At resubmission of your final files, please consider providing an ORCID ID for as many contributing authors as possible.

Please note that JCB requires authors to submit Source Data used to generate figures containing gels and Western blots with all revised manuscripts. This Source Data consists of fully uncropped and unprocessed images for each gel/blot displayed in the main and supplemental figures. Since your paper includes cropped gel and/or blot images, please be sure to provide one Source Data file for each figure that contains gels and/or blots along with your revised manuscript files. File names for Source Data figures should be alphanumeric without any spaces or special characters (i.e., SourceDataF#, where F# refers to the associated main figure number or SourceDataFS# for those associated with Supplementary figures). The lanes of the gels/blots should be labeled as they are in the associated figure, the place where cropping was applied should be marked (with a box), and molecular weight/size standards should be labeled wherever possible.

Journal of Cell Biology now requires a data availability statement for all research article submissions. These statements will be published in the article directly above the Acknowledgments. The statement should address all data underlying the research presented in the manuscript. Please visit the JCB instructions for authors for guidelines and examples of statements at (<https://rupress.org/jcb/pages/editorial-policies#data-availability-statement>).

B. FINAL FILES:

****It is JCB policy that if requested, original data images must be made available to the editors. Failure to provide original images upon request will result in unavoidable delays in publication. Please ensure that you have access to all original data images prior to final submission.****

****The license to publish form must be signed before your manuscript can be sent to production. A link to the electronic license to publish form will be sent to the corresponding author only. Please take a moment to check your funder requirements before choosing the appropriate license.****

Thank you for this interesting contribution, we look forward to publishing your paper in Journal of Cell Biology.

Sincerely,

Alex Mogilner
Monitoring Editor
Journal of Cell Biology

Tim Fessenden
Scientific Editor
Journal of Cell Biology

Reviewer #1 (Comments to the Authors (Required)):

The manuscript presents the results of a sophisticated biochemical and biophysical analysis of the yeast myosin 1, Myo5, that plays a critical role in clathrin-mediated endocytosis. This study establishes that Myo5, and presumably the closely related Myo3, is a power generating myosin and not a force-sensitive anchor. This leads the authors to propose that the role of these myosins in endocytosis is the generation of force that pulls actin filaments away from the membrane to allow the addition of new monomers to push the endocytic vesicle into the cell.

The authors have addressed the main comments by revising the text but they do not provide any in vivo experiments, instead they describe previously published work (which consists of a large body of detailed work). The text now better highlights experimental data available in the literature that provides support for their conclusion that yeast Myo5 acts as a force generator. Specifically, work from the Kaksonen group (Manenschijn, 2019) showing that the internalization speed of endocytic vesicles varies with numbers of Myo1s (Myo5 and Myo3 combined). Most notably, a 2-fold excess of Myo5 motors at the site of endocytosis results in faster internalization. While the authors reasonably interpret the findings reported in Manenschijn et al. to be consistent with Myo5 being a force insensitive motor (lines 319-322) it is not stated whether a force sensitive motor would give the same result if increased in number.

Reviewer #2 (Comments to the Authors (Required)):

The authors have satisfactorily addressed most of my concerns.

Reviewer #3 (Comments to the Authors (Required)):

I am satisfied by the revisions the authors have made.